# Reduced Endocannabinoid Tone in Saliva of Chronic Orofacial Pain Patients

**DOI:** 10.3390/molecules27144662

**Published:** 2022-07-21

**Authors:** Yaron Haviv, Olga Georgiev, Tal Gaver-Bracha, Sharleen Hamad, Alina Nemirovski, Rivka Hadar, Yair Sharav, Doron J. Aframian, Yariv Brotman, Joseph Tam

**Affiliations:** 1Department of Oral Medicine, Sedation and Maxillofacial Imaging, Hebrew University-Hadassah School of Dental Medicine, Jerusalem 91120, Israel; oly.geo@gmail.com (O.G.); dr.talgb@gmail.com (T.G.-B.); sharavy@mail.huji.ac.il (Y.S.); dorona@hadassah.org.il (D.J.A.); yossit@ekmd.huji.ac.il (J.T.); 2In Partial Fulfillment of DMD Requirements, Hebrew University-Hadassah School of Dental Medicine, Jerusalem 91120, Israel; 3Obesity and Metabolism Laboratory, The Institute for Drug Research, School of Pharmacy, Faculty of Medicine, The Hebrew University of Jerusalem, Jerusalem 9112001, Israel; sharleen.hamad92@gmail.com (S.H.); alina.nemirovskai@mail.huji.ac.il (A.N.); rivkaha58@gmail.com (R.H.); 4Department of Life Sciences, Ben-Gurion University of the Negev, Beersheba 8410501, Israel; brotmany@post.bgu.ac.il

**Keywords:** endocannabinoids, anandamide, 2-AG, chronic pain, orofacial pain, neuropathic pain, migraine, saliva

## Abstract

Background: the endocannabinoid system (ECS) participates in many physiological and pathological processes including pain generation, modulation, and sensation. Its involvement in chronic orofacial pain (OFP) in general, and the reflection of its involvement in OFP in salivary endocannabinoid (eCBs) levels in particular, has not been examined. Objectives: to evaluate the association between salivary (eCBs) levels and chronic OFP. Methods: salivary levels of 2 eCBs, anandamide (AEA), 2-arachidonoylglycerol (2-AG), 2 endocannabinoid-like compounds*N*-palmitoylethanolamine (PEA), *N*-oleoylethanolamine (OEA), and their endogenous precursor and breakdown product, arachidonic acid (AA), were analyzed using liquid chromatography/tandem mass spectrometry in 83 chronic OFP patients and 43 pain-free controls. The chronic OFP patients were divided according to diagnosis into musculoskeletal, neurovascular/migraine, and neuropathic pain types. Results: chronic OFP patients had lower levels of OEA (*p* = 0.02) and 2-AG *(p* = 0.01). Analyzing specific pain types revealed lower levels of AEA and OEA in the neurovascular group *(p =* 0.04, 0.02, respectively), and 2-AG in the neuropathic group compared to controls (*p =* 0.05). No significant differences were found between the musculoskeletal pain group and controls. Higher pain intensity was accompanied by lower levels of AA (*p* = 0.028), in neuropathic group. Conclusions: lower levels of eCBs were found in the saliva of chronic OFP patients compared to controls, specifically those with neurovascular/migraine, and neuropathic pain. The detection of changes in salivary endocannabinoids levels related to OFP adds a new dimension to our understanding of OFP mechanisms, and may have diagnostic as well as therapeutic implications for pain.

## 1. Introduction

Endocannabinoids (eCBs) are major constituents of the endocannabinoid system (ECS), and include fatty acid derivatives such as *N*-arachidonoylethanolamine [anandamide (AEA)] and 2-arachidonoylglycerol (2-AG). Both ligands have a high affinity to cannabinoid-1 and 2 receptors (CB_1_R and CB_2_R, respectively), which are also activated by Δ^9^-tetrahydrocannabinol (Δ^9^-THC). This psychoactive component found in marijuana has pain reducing effects [1]. CB_1_Rs are primarily located at presynaptic sites in the nervous systems. CB_2_Rs are found predominantly (but not exclusively) on immune cells [2]. The ECS is important in many physiological and pathological pathways as well as acting as a homeostatic system, including pain regulation in general [3,4,5,6], and specifically neuropathic and inflammatory pain [7].

The role of the ECS in pain has been extensively studied. Cannabinoid receptors and ligands are found at most levels of the pain pathway from peripheral sites (peripheral nerves and immune cells) to central integration sites (spinal cord), and higher brain regions (periaqueductal grey and rostral ventrolateral medulla associated with descending control of pain) [8]. CB_1_Rs are also found in brain areas associated with pain sensation (thalamus and amygdala) and body sites associated with pain processing and transfer [9]. The peripheral location CB_2_Rs enables involvement in pain mechanisms by preventing pro-inflammatory cytokine secretion near nerve endings [10]. Furthermore, eCBs may inhibit pain processes related neurotransmitter secretion via activation of cannabinoid receptors [11] and may also stimulate thermosensitive transient receptor potential (TRP) channels, such as TRPV1, known to be involved in pain stimulation.

In addition to the main eCBs, several eCB-like molecules have been identified and characterized such as *N*-palmitoylethanolamide (PEA) and *N*-oleoylethanolamide (OEA), which do not bind to the cannabinoid receptors, but have the same degradation pathway as AEA involving the membrane-associated fatty-acid amide hydrolase (FAAH), which catabolizes these ligands into free arachidonic acid (for AEA), palmitic (for PEA), and oleic (for OEA) acids and ethanolamine. The endogenous levels of these ligands were found to be modulated in pain and inflammatory murine models, suggesting eCBs play a role in the tonic inhibition of pain responses and the setting of nociceptive thresholds [12]. In this study (for simplicity reasons) these components also will be termed eCBs as AEA and 2-AG. Interestingly, changes in eCB ‘tone’, represented by the expression of cannabinoid receptors, their functional activity (upregulated or downregulated), and the relative amount of eCBs, may render a subject susceptible to disease. In fact, reduced eCB ‘tone’, also termed ‘clinical eCB deficiency syndrome’, is associated with migraine, fibromyalgia, irritable bowel syndrome, schizophrenia, multiple sclerosis, Huntington’s, Parkinson’s, anorexia, chronic motion sickness, and autism [13,14,15]. Specifically, AEA levels were lower in the spinal fluid of patients with chronic headache such as migraine [5], and FAAH was suggested as a target for treatment in cases of AEA deficiency migraine [10].

Over the last decade saliva has been recognized as a “diagnostic window to the body” [16], because its gene product distribution across Gene Ontological categories, such as molecular function, biological processes, and cellular components, are very similar to plasma [17]. In most studies regarding pain biomarkers, plasma has been the analytical medium and only a handful, have assessed the saliva concentration of these pain markers [18,19,20,21]; one of them has been recently published by our group [22]. Saliva has many advantages over serum as a diagnostic fluid, it is easy to collect, store and ship and can be obtained at a low cost in sufficient quantities for analysis. For patients, the non-invasive collecting techniques dramatically reduce anxiety and discomfort and simplify procurement of repeated samples for longitudinal monitoring. Saliva collection is safer than venipuncture, which exposes health care providers to infectious diseases. Saliva is also easier to handle for diagnostic procedures since it does not clot, reducing the manipulations required [23].

In fact, components of the ECS were found in saliva in conditions such as stress, fear, obesity, and malnutrition [24,25,26,27]. However, as far as we know, no data regarding salivary eCB levels in chronic pain patients in general and those with craniofacial pain in particular, have been collected or examined. Importantly, by evaluating salivary eCBs levels we may achieve a breakthrough in diagnosis and treatment individualization, such as fine-tuning when medical cannabis, a valid therapeutic agent for treating pain, would be beneficial.

## 2. Methods

### 2.1. Participants

The study followed the STROBE guidelines and met the requirements of the Hadassah Medical Organization’s (HMO) Institutional Review Board (IRB) (Approval No. 0662-HMO-17). All data were fully anonymized; Informed consents were waived according to the instructions of the Ethics Committee. The medical records of 83 (56 female, 27 male) orofacial pain (OFP) patients attending the Orofacial Pain Clinic, at the Hebrew University-Hadassah Faculty of Dental Medicine, between 2017–2018 were reviewed. 43 age-matched (28 female, 15 male) pain-free participants acted as a control group. Inclusion criteria: over 18 years of age; diagnosis of chronic OFP for at least 3 months; able to provide saliva sample of at least 200 µL per 10 min. Exclusion criteria: cannabis use, refusal or inability to consent, medical conditions affecting salivary gland function such as autoimmune disease primarily Sjogren’s syndrome and rheumatoid arthritis, history of head and neck cancer treated using radiotherapy, chemotherapy and/or biological targeted therapies, graft versus host disease, fibromyalgia.

### 2.2. Orofacial Pain Diagnosis

Based on OFP diagnosis, etiology and pain characteristics the patients were divided into 3 groups as described by Sharav and Benoliel [28]:

*Musculoskeletal group*: Pain from temporomandibular disorders (TMD) including masticatory muscle pain (MMP), temporomandibular joint (TMJ) pain, and combined muscle and joint pain. Diagnosed according to the Diagnostic Criteria for Temporomandibular Disorders (DC/TMD) [29].

*Neurovascular group*: Migraine and Tension type headache, diagnosed according to the ICHD-3 [30] and “neurovascular orofacial pain” (NVOP) for facial pain with migrainous features in the second and/or third divisions of the trigeminal nerve [31].

*Neuropathic group:* Trigeminal neuralgia (TN) [32], Painful post-traumatic trigeminal neuropathy (PTN) [33], Burning mouth syndrome (BMS), post herpetic neuralgia (PHN), and persistent idiopathic facial pain (PIFP) [34].

The thorough extra oral examination included: cranial nerve examination and masticatory apparatus palpation as previously described [35,36]. Intra-oral examination was performed to exclude dental, periodontal and mucosal pathology. Brain and brainstem imaging were performed for TN to exclude intracranial pathology.

### 2.3. Collection of Medical Records

Demographic data including gender, age, BMI, medications, and relevant medical history was collected. Pain characteristics including onset (months), intensity, and quality were recorded on the intake form routinely used in our clinic [32,36,37]. Pain intensity was rated using a verbal pain scale (VPS) in the saliva collection day, where 0 represents no pain and 10 the worst imaginable pain.

### 2.4. Saliva Collection

Unstimulated saliva was collected for 10 min as described previously [22,38] into pre-calibrated tubes. Saliva was collected between 9:00 AM and 12:00 PM. All participants refrained from exercising, eating, drinking and brushing their teeth in the 2 h before saliva collection. Patients did not take their medications, including sialagogues, before saliva collection. Volunteers rested for 10 min before saliva collection, sitting in an upright position in a quiet room and were asked not to speak or leave the room until after the saliva was collected.

Saliva samples were immediately stored at −80 °C, until further processing. After defrosting, the samples were centrifuged at 3500× *g* for 10 min at 2 °C to remove insoluble materials, cell debris and food remnants. The supernatant was aliquoted into polypropylene tubes and then stored at −80 °C until further processing. 

The samples were run in the following order, 2 OFP samples from the same group and then a gender- and BMI-matched control sample.

Endocannabinoid extraction, purification, and measurement:

The extraction, purification, and quantification of saliva eCBs was performed by stable isotope dilution liquid chromatography/tandem mass spectrometry (LC-MS/MS) as previously described by us [14,39,40,41,42]. Briefly, AEA, 2-AG, their breakdown product arachidonic acid (AA), PEA, and OEA were extracted, purified, and quantified as follows. Total salivary protein was precipitated using ice-cold acetone and Tris buffer (50 mM, pH 8.0). Samples were then homogenized using a mixture of 0.5 mL ice-cold methanol/Tris buffer (50 mM, pH 8.0), 1:1, and 7 µL internal standard (22.4 ng d_4_-AEA). The homogenates were then extracted using ice-cold CHCl_3_:MeOH (2:1, *v/v*), and immediately washed with ice-cold chloroform three times. The samples were dried under a thin stream of nitrogen and reconstituted in MeOH. Analysis by LC-MS was conducted using an AB Sciex (Framingham, MA, USA) Triple Quad 5500 Mass Spectrometer and a Shimadzu (Kyoto, Japan) UHPLC System. Liquid chromatographic separation was acquired using a Kinetex (Phenomenex) column (C18, 2.6 mm particle size, 100 * 2.1 mm). The levels of each compound were analyzed by multiple reaction monitoring. The molecular ion and fragment for each compound were measured as follows: m/z 348.3 → 62.1 (quantifier) and 91.1 (qualifier) for AEA, m/z 379.3 → 91.1 (quantifier) and 287.3 (qualifier) for 2-AG, m/z 305.2 → 91.1 (quantifier) and 77.1 (qualifier) for AA, m/z 326.2 → 62.1 (quantifier) and 55.1 (qualifier) for OEA, m/z 300.3 → 283.2 (quantifier) and 62.0 (qualifier) for PEA, and m/z 352.3 → 66.1 (quantifier) and 91.1 (qualifier) for [^2^H_4_] AEA. Sample levels of AEA, 2-AG, AA, PEA, and OEA were measured against a standard curve and then expressed as fmol/mg proteins.

### 2.5. Statistics: The Statistical Analysis Was Performed Using SPSS Version 25

To examine the differences in eCB levels for nominal and categorical background variables, *t*-tests and one-way analysis of variance were performed, additional post-hoc Scheffe tests were performed when differences were significant. Spearman coordinator was used to examine the specific categories that made up the differences. The differences between eCBs types and specific background variables were examined for each diagnosis separately by using the non-parametric Mann Whitney and Kruskal–Wallis test.

## 3. Results

Of the 126 participants in the study, 83 had chronic OFP (Figure 1), 21 (25.4%) in the musculoskeletal group, 31 (37.3%) in the neurovascular group, and 31 (37.3%) in the neuropathic group (Figure 1A). The remaining 43 were the pain-free control group. The specific pain diagnoses and incidence of each diagnosis within general pain group are summarized in Figure 1C. Figure 2, summarizes gender distribution according to the different groups.

64 chronic OFP patients used pain medication including: Tricyclic antidepressants—Amitriptyline, Nortriptyline (27 patients), Pregabain (10 patients in the neuropathic group), Carbamazepine (10 in the neuropathic group), Topiramte (10 patients in the neurovascular group), Duloxetine (3 patients), Clonazepam (10 patients in the neuropathic group), Valporic Acid (1 patient in the neurovascular group), Tripans (1 patient, as abortive treatment in the neurovascular group). Some of the patients used more than one medication (for details see: Appendix A).

Mean salivary levels (fmol/mg protein) of eCBs: AEA, 2-AG, OEA, PEA, and AA are summarized in Table 1. Significantly higher levels of AA, OEA and PEA were found in men compared to women (*p* < 0.001) (Table 1). Specifically—OEA, PEA and AA levels were lower in women than men in the neurovascular group (*p* = 0.002, *p* < 0.001, *p* = 0.048, respectively), women had lower levels of 2-AG and AA than men in the neuropathic group (*p* = 0.061, *p* = 0.037, respectively)—data not shown.

Table 2 presents significant higher levels of OEA (*p* = 0.02), and 2-AG (*p* = 0.01) in the control group compared to the pain groups. The lower part of the table summarizes the differences in salivary eCBs in the three pain groups (musculoskeletal, neurovascular and neuropathic) compared to controls. 

Table 3 summarizes significant differences between each pain group and controls, specifically AEA and OEA were lower in neurovascular group (*p* = 0.02, 0.04, respectively), 2-AG was lower in the neuropathic group (*p* = 0.05). No significant difference was found between the musculoskeletal group and controls.

Table 4 summarizes salivary eCBs levels in relation to pain severity within neuropathic and neurovascular groups: In the neuropathic group, significant differences were found in AA levels (*p* = 0.028) between those with mild versus moderate and severe pain.

## 4. Discussion

In this study, we highlight, for the first time, the significant reduction in salivary eCB levels in chronic OFP patients. Consequently, we suggest salivary eCB levels may be utilized as biomarkers for OFP. The ECS is involved in a number of physiological and pathological processes. Its anti-nociceptive effects appear to be mediated by CB_1_Rs within the CNS [43,44] and via both CB_1_Rs and CB_2_Rs peripherally [45,46]. Pain sensation can be modulated by the ECS at receptor and ligand levels. At the receptor level, CB_1_R expression was lower in mice with neuropathic diabetes compared to controls [47]. Similarly, CB_2_R levels are lower in chronic versus acute pain patients [48]. In contrast, FAAH mRNA levels were higher in chronic pain patients, implying enhanced AEA degradation. At the ligand level, Stensson et al., demonstrated higher AEA and 2-AG levels in patients with fibromyalgia following 15 weeks of strength-based exercise which decreased pain and depression. The authors hypothesized that this was due to the effect of eCBs on CB_2_R. The effect was unrelated to the CNS level, as sensation and acute pain threshold were unchanged [49].

Data on cannabinoid treatment for headache and OFP are of poor quality and based on case series, case reports, anecdotal reports or retrospective analysis [50]. We hypothesized that the evaluation of levels of 5 salivary eCBs in chronic OFP patients may enhance our understanding of the diagnosis and treatment of these patients, and therefore we examined their salivary levels in 83 chronic OFP patients with different pain etiologies (musculoskeletal, neurovascular and neuropathic) and compared them to 43 pain-free controls. Overall salivary eCBs levels were lower in the chronic OFP patients than controls (Table 3), and these changes were significant for the 3 eCBs–2-AG, OEA, and AA. 2-AG influences pain sensation via multiple mechanisms, including its effects on TRPV1, CB_1_R, and CB_2_R signaling, and by serving as a prostaglandin precursor, thereby modulating inflammation and pain [51]. OEA levels were significantly reduced in our pain patients, however its role in pain modulation is unclear. Lower visceral and inflammatory responses via PPARα-mediated signaling in mice [52] suggests that OEA may act via TRPV1 [53]. OEA may have several physiological roles, including pain perception, but some studies suggest that it has a lesser impact on pain, and a greater effect on energy balance [54,55].

The neuropathic pain patients in the current study had significant reductions in salivary 2-AG levels. Indeed, decreased plasma 2-AG levels have been measured in neuropathic pain patients with optic neuromyelitis [56] Donvito and colleagues showed that mice without CB_1_R and CB_2_R experienced neuropathic pain, while overexpression of CB_2_R prevented pain [1]. There were no significant differences in salivary eCB levels in the musculoskeletal pain patients compared to controls, and this may be due to the less severe nature of the pain (especially when considering neurovascular or neuropathic pain) [28].

Consistent with this notion, we found that those with higher pain intensity had lower eCB levels (specifically AA; Table 4). The main and significant differences were found when comparing mild pain (VPS 1–3) to moderate (VPS 4–6) and severe pain (VPS 7–10). No significant difference was found between moderate and severe pain. These data suggest that eCB levels correlate with pain sensation, and that their salivary levels may be used as an aid in the diagnosis and assessment of pain. Additionally, salivary PEA, AA and OEA levels were significantly higher in men than women, as reported by Fanelli et al., regarding 2-AG [57]. These differences may be due to physiological differences in expression and localization of ECS components in males and females [58] and may partially explain the higher incidence of chronic pain in general and craniofacial pain in particular in women [28,59].

eCBs have been detected in the saliva under various circumstances, e.g., stress [25,60], food, taste, and appetite [27,61], as well as obesity [26]. Whereas our results do not imply that salivary eCBs are directly related to central eCBs, they rather suggest that they may be associated with orofacial pain (OFP). One may hypothesis that central eCB levels could be reflected by changes in eCB ‘tone’ in the blood and then may influence salivary content of eCBs, since salivary components originate from the salivary glands or may be derived from passive diffusion or active transport from blood that is filtered and processed from the rich glandular vasculature. However, this assumption should be experimentally validated in future studies.

Our findings also imply the existence of a faulty mechanism of pain modulation in chronic OFP patients, as previously suggested for other modalities, e.g., defective conditioned pain modulation [62,63,64]. The fundamental mechanism where pain is for survival, and therefore beneficial, is disrupted in chronic pain. This may be due to, or associated with, lower eCBs levels. Indeed, human studies include many factors that may influence the results, some of which are quantifiable, and others that are seemingly impossible to measure. This is even more apparent when studying pain, an experience involving emotions and sensations. For example, prophylactic and abortive pain medications may affect results, and they have diverse influences and side effects for different patients with different diagnoses. Psychological elements can be primary or secondary to chronic pain, and these elements are highly individual, and may be related to medication dosage, absorption, sleep quality, and even to the queue at the entrance to the clinic. In this case, eCBs concentrations are dynamic and depends on circadian rhythm (particularly 2-AG), stress, inflammation and others [65]. Any of these factors can influence pain and eCBs levels directly or indirectly.

Taken together, a more focused, selective study is required to: (i) confirm the accuracy of these results for specific pain diagnoses and at different time points; (ii) investigate the effects of successful pain treatment on eCB levels; (iii) identify high-risk groups in order to enable early diagnosis, screening and therapeutic follow-up; and (iv) determine if salivary eCB levels are indeed related to central eCB ‘tone’.

In conclusion, the findings of the current study, that salivary eCBs levels are lower in chronic pain patients than in pain-free individuals, may help select the best therapeutic interventions for these patients. For instance, specific medications or formulations may alter the enzymes involved in the breakdown/synthesis of eCBs, and indirectly affect eCB ‘tone’. Moreover, salivary eCB levels can be used as a diagnostic tool and predictor of success before medical cannabis is used in this patient population.

## Figures and Tables

**Figure 1 molecules-27-04662-f001:**
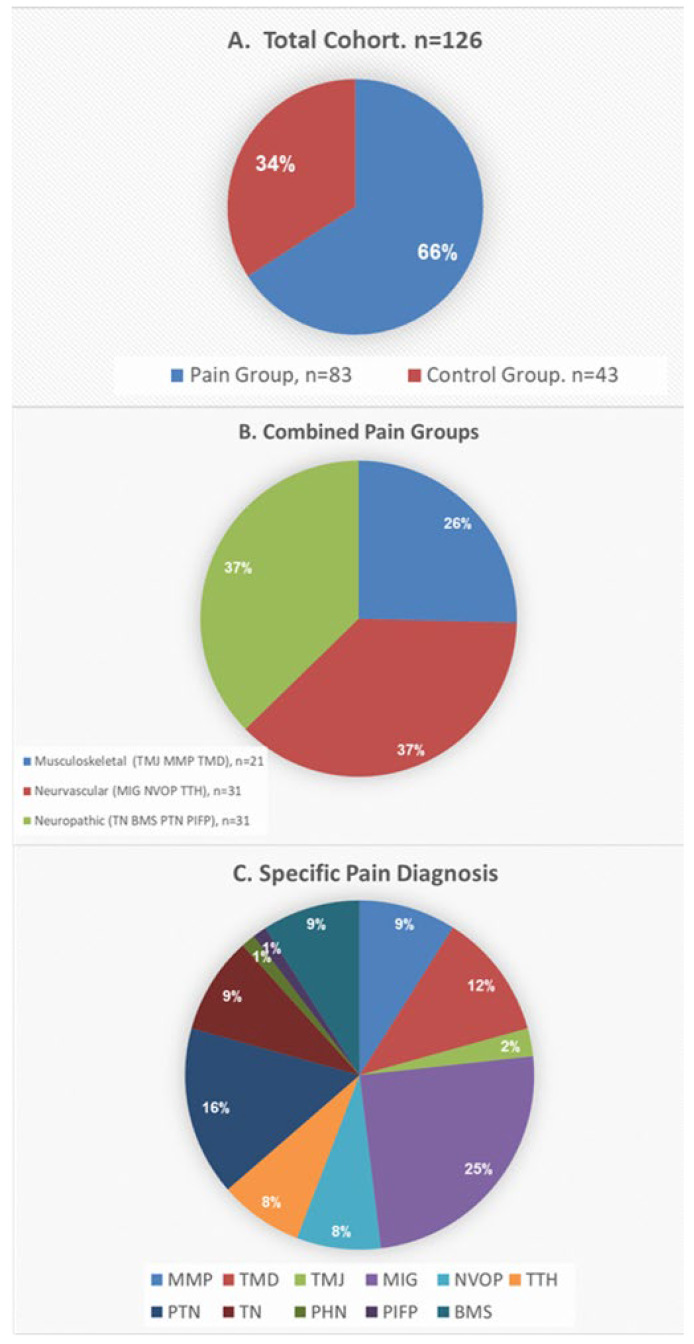
Cohort distribution according to diagnosis. Diagnoses are grouped according to etiology and characteristics, see Sharav and Benoliel [28]: MMP—Masticatory Myofascial Pain, TMD—Temporomandibular disorder, TMJ—Solely Temporomandibular joint origin, Mig—Migraine, NVOP—Neurovascular orofacial pain (orofacial migraine), TTH—Tension Type Headache (primary headache), PTN—Post Traumatic Neuropathy, TN—Trigeminal neuralgia, PHN—Post Herpetic Neuralgia, PIFP—Persistent idiopathic facial pain, BMS—Burning Mouth Syndrome.

**Figure 2 molecules-27-04662-f002:**
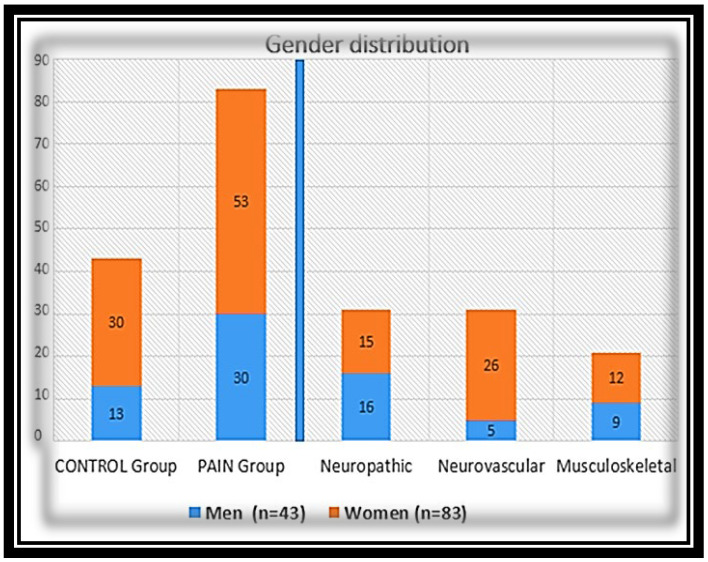
Cohort distribution according to gender.

**Table 1 molecules-27-04662-t001:** Mean salivary eCBs levels in the total cohort (126 patients) and in men and women (*fmol/mg protein*).

Endocannabinoids (eCBs)	fmol/mg (Mean ± SD)	Women (Mean ± SD)	Men (Mean ± SD)	*p* Value
**AEA**	0.17 ± 0.3	0.15 ± 0.17	0.21 ± 0.69	0.28
**2-AG**	41.3 ± 40.6	44.15 ± 45.32	37.71 ± 29.25	0.40
**OEA**	49.1 ± 59.8	37.93 ± 50.35	73.16 ± 70.33	**<0.001**
**PEA**	11.5 ± 11.8	8.34 ± 7.96	17.73 ± 15.30	**<0.001**
**AA**	7659.1 ± 1876.1	12112.25 ± 1154	3262.18 ± 2452	**<0.001**

2-AG -2-arachidonoylglycerol, PEA—*N*-palmitoylethanolamine, AEA—*N*-arachidonoylethanolamine, AA—arachidonic acid, OEA—*N*-oleoylethanolamine. -*t*-test.

**Table 2 molecules-27-04662-t002:** Salivary eCBS levels in controls compared to total pain group and specific pain groups *(fmol/mg protein)*.

*** Group**	**AA** **(Mean ± SD)**	**PEA** **(Mean ± SD)**	**OEA** **(Mean ± SD)**	**2-AG** **(Mean ± SD)**	**AEA** **(Mean ± SD)**
**Control**	2327.1 ± 2539	12.9 ± 11.75	67.7 ± 77.09	54.71 ± 36.47	0.2 ± 0.18
**Pain**	1710.2 ± 1552	10.7 ± 11.93	41.2 ± 47.04	35.88 ± 41.29	0.17 ± 0.2
***p*-value **	0.09	0.34	**0.02**	**0.01**	0.94
Musculoskeletal	2177.3 ± 1868	9.9 ± 9.30	40.4 ± 37.06	39.6 ± 40.91	0.1 ± 0.14
Neurovascular	1354.3 ± 1354	8.8 ± 10.21	35.6 ± 40.31	33.2 ± 41.45	0.0 ± 0.09
Neuropathic	1749.80 ± 143	13.2 ± 14.71	47.3 ± 58.63	35.9 ± 42.56	0.2 ± 0.45
*p*-value	0.17	0.36	0.11	0.09	0.07

* Neuropathic group: BMS, TN, PHN, PIFP. Neurovascular group: primary headaches—MIG, NVOP, TTH. Musculoskeletal group: MMP, solely TMJ, TMD (combined MMP and TMJ). The data was subjected to *t*-test for Control vs. Pain and One way AVOVA for Control vs. specific pain groups.

**Table 3 molecules-27-04662-t003:** Salivary eCBs in specific pain groups compared to controls (*fmol/mg protein*).

Group	eCBs	Pain (mean ± SD)	Control (mean ± SD)	*p*-Value
**Neurovascular**	**AEA**	0.09 ± 0.09	0.17 ± 0.18	**0.02**
	**OEA**	35.65 ± 40.31	67.7 ± 77.09	**0.04**
	**AA**	1354.3 ± 1386.94	2327.15 ± 2539.27	0.06
**Neuropathic**	**2-AG**	35.97 ± 42.56	54.71 ± 36.47	**0.05**
**Musculoskeletal**	**None**	-	-	NS

- The data was subjected to non-parametric Mann-Whitney *U* test.

**Table 4 molecules-27-04662-t004:** eCBs in relation to pain severity levels in neuropathic and neurovascular pain groups (*fmol/mg protein*).

Group	eCBs	Pain Level *	*n*	fmol/mg Protein (Mean ± SD)	*p*-Value
Neurovascular	**OEA**	Mild	3	85.42 ± 97.27	0.097
		Moderate	7	30.83 ± 24.87	
		Severe	19	31.65 ± 30.98	
	**PEA**	Mild	2	22.44 ± 27.85	0.062
		Moderate	7	7.97 ± 6.86	
		Severe	14	7.64 ± 5.86	
Neuropathic	**PEA**	Mild	3	17.26 ± 9.96	0.067
		Moderate	7	5.57 ± 4.44	
		Severe	14	9.02 ± 7.14	
	**AA**	Mild	2	4027.43 ± 596	**0.028**
		Moderate	7	1214.79 ± 1336	
		Severe	14	1514.9 ± 1202	

* Pain level, according to VPS (Verbal Pain Score). Mild = VPS:1–3, Moderate = VPS:4–6, Severe = VPS:7–10./Nonparametric Kruskal–Wallis test.

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
