# Peer review of "Reduced Endocannabinoid Tone in Saliva of Chronic Orofacial Pain Patients"

_molecules, 2022, doi:10.3390/molecules27144662_

Round 1

Reviewer 1 Report

Overview. The authors present novel data indicating the differences in salivary endocannabinoids (AEA & 2-AG), lipid mediators (OEA & PEA), and product of eCB metabolism (AA). Results indicate sex differences in salivary lipid content and an association of lower lipid content with pain incidence and severity.

Major concerns. Authors indicate a significant difference between male and female participants in the total cohort, however sex differences for control and pain groups separately are not shown. As well, this major difference may skew subsequent analyses as the distribution of sexes is not indicated for combined or specific pain groups. This separation of a major covariate is crucial to the understanding of these results and their interpretation moving forward. As well, these associations strongly suggest an interplay between pain and salivary eCB content, however this study would be strongly supported by comparisons of salivary lipids to established sources (e.g., plasma, CSF, etc.). Lastly, specific statistical analyses and their interpretation should be elaborate either in footnotes of table or in methods to indicate context for each analysis.

Specific comments below:

Sentence starting on line 58, “The peripheral location…” should be reworded for clarity.

Table 1. Please modify text formatting for consistency. As well, manuscript text indicates that AA levels are lower in women, however text in the table indicates AA levels in women are 12112.25 fmol/mg compared to men with 3262.18, as well the SD for AA are in the thousands. These confounding values may be due to typo or erroneous loss of decimal point. Please include corrected values or update text correspondingly.

Line 217, it is not appropriate to describe values as almost significant. This reviewer suggests reordering the metabolites to show significant results (i.e., OEA and 2-AG in this context) first in text. As well, double check for grammar and syntax, “...the control group compared the pain groups.” (Line 218) is missing the word “to”.

Table 2. Please indicate whether errors are SD or SEM. These values are very large, sometimes being greater than the mean. Again, it is not appropriate to indicate results as “close to significance.”

Table 3. Similar to table 2, indicate the error type. It may also be beneficial to include the specific statistical test being used in each table of values, as methods sections does not indicate in which contexts the specific tests were used. Given the study design and large error values, it may also be important to compare confidence intervals for statistical analysis.

Table 3. Why are eCB levels not all indicated for each group. Despite not showing significant change, it may be important to indicate levels of eCBs detected.

Line 236. AA is indicated twice with different P-values, as well the P-values in the text do not match the data described in table 4. This is a large discrepancy that needs to be resolved.

Table 4. What does P-value represent? Are these the results of ANOVA or specific comparisons between groups?

In the first para of discussion, changes to the ECS in CNS during neuropathy and fibromyalgia are mentioned, however the site of these changes and relevance to the current study are not readily apparent. How were these findings measured and what is their bearing on the current investigation?

Author Response

Please see the attachment:
letter for  Reviewer 1

Reviewer 2 Report

The authors evaluated the eCB salivary levels of three groups of OFP patients comparing them to controls. Interestingly they found that eCB levels are correlated with pain severity. 

LINE 189 Please write here Figure 1,B

LINE 218-222 the explanation is not clear. I did not found p=0.07

LINE 279 Please write inhibitor of FAAH and MAGL instead of inhibitor of AEA and 2-AG

LINE 439 the second author is missing

Supplementary please check the spelling of pain groups

Author Response

Please see the attachment:
letter for  Reviewer 2

Reviewer 3 Report

The MS molecules1790838 present that salivary eCBs levels are lower in chronic orofacial pain patients than in pain-free individuals and that salivary eCB levels may be utilized as biomarkers for orofacial pain. 

Although, this hypothesis sounds interesting, I have few comments concerning the MS: 

I suggest to modify the title, as mainly PEA and AA level were changed, but not classical endocannabinoids.  

Number of sample in the table 1-3 will help to analize obtained results and follow the statistic.  

Discussion is too long, the proposed anti-nociceptive mechanism of determided molecules are not examined in this study, so could be ommited. 

Authors described that any of medications taken by patients have not affected the results. The Authors should describe in detail if medication/s was/were withdrown before saliva sample collection. If so, for how long. It is interesting if dosage and frequency of treatment was suffiecient to relief pain and if it has an effect on „endocannabinoids” level obtained? 

Limitation of the study should be also mentioned, i.e. in the context  that the endocannabinoids are present in the human fluids and their concentrations are dynamic and depends on circadian rhythm (particularly 2-AG), stress, inflammation and others (see Hillard CJ. Neuropsychopharmacology. 2018 Jan;43(1):155-172. doi: 10.1038/npp.2017.130). Could they have an impact on the results obtained in this study?

Author Response

Please see the attachment:
letter for  Reviewer 3

Round 2

Reviewer 1 Report

I thank the authors for their thoughtful responses and the tremendous efforts in the study herein.